# Cutting Tests of the Outer Layer of Material Using Onion as an Example

**DOI:** 10.3390/ma14092360

**Published:** 2021-05-01

**Authors:** Andrzej Bochat, Marcin Zastempowski, Marcin Wachowicz

**Affiliations:** 1Faculty of Mechanical Engineering, UTP University of Science and Technology Bydgoszcz, Al. prof. S. Kaliskiego 7, 85-796 Bydgoszcz, Poland; bochat@utp.edu.pl; 2Metal Progress, Krusza Duchowna 30, 88-100 Krusza Duchowna, Poland; metalprogress@poczta.fm

**Keywords:** cutting plant material, machine cutting efficiency, onion peeling efficiency, new machine design, onion peeling, modular machine

## Abstract

This paper describes experimental research on cutting the outer layer of onions in the machine peeling process. The authors’ own globally innovative modular machine construction was used for this purpose. The onion peeling machine was constructed on a real scale. The effectiveness of the machine’s functioning $S_{e}$ was defined as the ratio of the mass of material correctly removed by the scale blower $m_{p}$ to the mass of all material leaving the machine on the test bench $m_{c}$. In order to carry out the experimental research, a test stand was constructed, a research plan and programme were adopted, and the research methodology was developed. The results obtained during the experimental research and the data obtained from the regression function equations for the developed design of the onion peeling machine were used to build systems of independent variables, for which the dependent variable $S_{e}$ reached extreme values. The effectiveness of the machine’s operation $S_{e}$ of modular construction increased with the increase in the depth of the external incisions of the shells $d_{n}$, the number of scale-blowing nozzles, and the pressure of the air supply to the scale-blowing unit p. Increasing the material feed rate $v_{p}$ and the distance of the air nozzles from the material to be processed $h_{d}$ reduced the machine’s efficiency $S_{e}$. The tests carried out showed a high level of efficiency on the level of $S_{e}=0.645-0.780$, which is not found in mass-produced machines.

## 1. Introduction

Edible onions have important nutritional and medicinal properties [1,2,3]. Nowadays, the vast majority of these harvested vegetables is processed industrially as ingredients in canned and frozen foods, salads, and as semi-finished products used in related areas of the food industry to ensure specific taste requirements are met. In recent years, the world’s production of onions has averaged between 85 and 88 million tonnes annually, requiring a cultivated area of 4.4 million hectares [4]. Geographically, Asia is the largest producer of onions, while Europe accounts for more than 15% of the world’s production. All varieties of onions promote health, act as antibacterial agents, and improve the immune system. Moreover, onions have anti-inflammatory and expectorant effects. They are an ideal remedy for all varieties of upper respiratory tract infections, and are also used to cure wounds, burns, and scalds.

Irrespective of their subsequent use, the first treatments an onion undergoes after being harvested are the removal of the outer, dried layers of scales and the removal of the remaining chives and stigmas containing the root. Despite the ongoing development of food-processing techniques, the abovementioned operations are still largely carried out manually. The main reasons for the lack of complete mechanisation of the onion peeling process include its high energy consumption, unsatisfactory efficiency, and higher waste generation compared with manual peeling. The described situation indicates a strong need to identify the process of onion peeling, and to introduce machines to the market for its mechanisation while ensuring the required high quality of the final processed material.

The leading manufacturers of onion peeling machines are, among others, Chinese manufacturers (Zhucheng City Dayang Food Machinery Co., Henan Gelgoog Machinery, Zhengzhou Weihe Machinery Trading Co., and Qingdao Xiaodao Food Machinery Co.), the USA (CMI Equipment & Engineering Co. and Angli (M&P), and the European Union (Horus, DxD, Imizumi, Projekt, Dofra, APH Group, Sormac, and Finis).

The existing constructional solutions of machines for peeling and cutting materials such as onions operate in three systems: cutting the onion scales and blowing them off without cutting the ends; cutting the onion scales, blowing them off, and then cutting the ends off; and a third system consisting of first cutting off the ends of the material and then cutting and blowing off the scales.

Despite the existence of onion-peeling machines on the food industry market, in the available literature, there is a lack of information concerning conducted research on the existing constructional solutions. Issues related to the cutting and crushing of plant material have been widely described [5,6,7,8,9,10,11,12,13]. Conversely, available scientific studies in the literature are lacking concerning vegetable cutting, especially related to the mechanisation of onion peeling processes, except for the machine description [14,15,16]. The producers also fail to provide any detailed information on the efficiency of the onion-peeling machines, i.e., they have not conducted detailed studies or published the results in the available literature. The generally available information relates only to the method of feeding the onions (manual or automatic) and the efficiency of the process.

Due to a noticeable lack of scientific papers on the processes of mechanised peeling of onions, it is not possible to unequivocally determine which characteristics and construction parameters of the machine have a decisive impact on the effectiveness of the process. The aim of the study was to solve the aforementioned research problems in this field and to answer the formulated research question concerning the influence of the selected design parameters of a new onion peeling machine on the effectiveness of its operation.

## 2. Materials and Methods

The effectiveness of the machine’s operation $S_{e}$ was defined as the ratio of the mass of onions peeled correctly by the scale-blower $m_{p}$ to the mass of all onions leaving the machine on the test bench $m_{c}$. Therefore, the effectiveness of the machine’s operation was calculated by the relation:(1)$$S_{e}=\frac{m_{p}}{m_{c}}$$

In order to experimentally verify the influence of the selected constructional parameters of the machine on the effectiveness of its operation, the plan and programme of the experiment were adopted, a self-designed research station with the research apparatus was constructed, the research methodology was developed, the experimental tests were carried out, and the method for the analysis of the research results was developed. A widely grown onion variety called Bonus was used as the research material. Figure 1 presents the material subjected to tests.

In order to achieve the purpose of the study, which was to answer the formulated research question, experimental research was planned. The following were assumed as the independent variables in the experimental research:material feed rate *v_p_*,depth of external shell incisions *d_n_*,number of air nozzles for de-scaling *n_d_*,distance of the air nozzles from the material *h_d_*, andpressure of the air supply to the de-scaling assembly *p*.

The following was assumed to be the dependent variable:effectiveness of the machine’s operation *S_e_*.

The independent variables adopted in the experiment were defined as follows:The material feeding speed *v_p_* was defined as the linear speed of the conveyor delivering the onion to the end-cutting zone and cutting the outer layer. This speed was converted into a unit (m·s^−1^) on the basis of the pre-set rotational speed of the propulsion engine and the values of the transmission ratios. However, in the presentation and elaboration of the results, this amount is provided in units of pieces/min.The depth of cuts in the outer layer (scales) *d_n_* (mm) were determined as the depth of penetration of the material-cutting knives; this can be adjusted by changing the position of the knife penetration stops.The number of air nozzles removing the cut material (scales) *n_d_* (pcs.) defines the number of active nozzles involved in the process of removing the scales from the onion with compressed air; the design of the machine allows for the use of one to four air nozzles.The distance of the air nozzles from the material *d_n_* (mm) was defined as the distance between their tips and the surface of the rotating and conveying rollers, which, after considering the diameter of the material to be cut, indirectly determines the distance of the nozzle tip from the surface of the onion.The pressure of the air that supplies the shell-removal system *p* (bar) was measured as the air pressure at the entrance to the pneumatic system.

The independent variables were defined on the basis of a thorough analysis of the literature in the field of mechanical and materials engineering from the perspective of the theory and design of machines, and on preliminary research carried out by the authors of this paper.

The values of independent variables that were adopted in the experimental studies are presented in Table 1.

During the experimental research, the cutting process was carried out on the material of the onion variety Bonus with the diameter of the onions falling within the range of 50 to 70 mm. The onions were stored in a warehouse with a relative air humidity of 80% to 95% and an ambient temperature of 10 to 15 °C. The experiment was designed according to the five-factor cross classification 3 × 3 × 3 × 4.

During the main experiments, 10 replications in the study sample were used on the basis of the preliminary studies and their statistical evaluation, with an assumed significance level α of 0.05 and an acceptable estimation error d of 2%.

In order to carry out fundamental experimental research based on the literature studies, an original test stand was constructed, which was designed to cut the ends (root and chive) of onions and to cut the surface layer in order to peel them from the scales [17].

Figure 2 depicts the test bench, and Figure 3 presents its block diagram as a modular-functional scheme.

The onions should be positioned in the carriers with the correct spatial orientation. The symmetry of the onions should be oriented horizontally and perpendicular to the direction of movement of the take-up conveyor shown in Figure 4. Once the onions are positioned in the carriers, they are transported to the end cutting area (Figure 5).

The end-cutting system consists of two identical disc knives (Figure 6) of 200 mm diameter, rotating at a speed of 2000 rpm, and driven by a three-phase induction motor. These knives are fixed in a manner that allows them to be continuously spaced and are connected with an apparatus that copies the size of the onion. The operation of this system consists of ensuring that the spacing between the cutting knives is automatically adjusted to the size of the onion. In order to ensure the constant position of the material during the cutting process, the onion is pressed to the carriers by a pressure system equipped with a synthetic belt, the linear speed of which is equal to the speed of the onion take-up conveyor.

For cutting the outer layer of the material (onion scales), a pair of knives operating in the horizontal plane and a pair of knives operating in the vertical plane were used. The geometrical form of the knives for cutting the scale in the horizontal and vertical planes together with the cutting system are presented in Figure 7.

Once the top layer of the onion material was incised, it was removed by the process of blowing off the waste (scales) with the use of compressed air and GreenTec MQL 47,003 nozzles. When this process was complete, the material was ready for packing. A view of the cleaning system is shown in the Figure 8.

The above-described modular machine for peeling onions that we developed has the following features distinguishing it from the machines available on the market:The knives for peeling the onion ends are mounted in a way that allows continuous change of their spacing, which, in combination with the copying system, allows changing the cutting line depending on the size of the material. Consequently, the material does not require any initial, manual selection and is fed automatically from the hopper.The scales are cut by one or two pairs of knives. Their activation depends on the type of the material to be cut in order to ensure the required quality of the final product.The system of blowing off the cut surface layer elements (scales) was designed with the possibility of adjustment and adaptation by changing the distance of nozzles from the cut material, as well as the number of active nozzles. In addition, an oscillating movement of the nozzle assembly was applied, which enables more efficient operation of the cleaning unit.The machine is equipped with an optical system manufactured by Sick, which enables the assessment of peeling efficiency and material cleanliness.

## 3. Results

In order to determine the influence of the selected design parameters of the onion-peeling machine on the effectiveness of its operation $S_{e}$, tests were performed on a test bench. The results of the experimental research were used to calculate the arithmetic mean of the dependent variable $S_{e}$. The arithmetic averages of the tested values obtained from the specific research points were determined statistically from sets of points with a small scatter of results. The ranges of values of selected statistics for the dependent variable $S_{e}$ were: standard deviation б = 0.018–0.027 and coefficient of variation $x_{z}$ = 0.02–0.03.

In order to further analyse the experimental results, a statistical analysis was carried out and, to this end, there was developed a multivariate regression function equation of the form:$$S_{e}=a_{1}+a_{2}v_{p}+a_{3}d_{n}+a_{4}n_{d}+a_{5}h_{d}+a_{6}p+a_{7}v_{p}^{2}+a_{8}d_{n}^{2}+a_{9}n_{d}^{2}+a_{10}h_{d}^{2}+a_{11}p^{2}$$
where $S_{e}$ is the generalised dependent variable obtained as a result of the experimental studies, $v_{p}$ is the feeding speed of the material, $d_{n}$ is the depth of incision of the scales, $n_{d}$ is the number of air nozzles, is the distance of air nozzles from the material, *p* is the pressure of the air supply to the air nozzles, and $a_{i}$ is the regression coefficients for i = 1–11.

The procedure for determining the regression function was performed numerically in the following steps:All 11 regression coefficients were determined and their significance was analysed at the significance level α = 0.05 by determining the *t_kr_* value from the Student’s *t* distribution for 324-11-1 degrees of freedom,Expressions containing structural parameters not statistically significant were eliminated,The regression equation was modified by the least statistically significant terms and the analysis of the regression coefficients was repeated until all the structural parameters of the equation satisfied the significance condition,The final equation of the multivariate regression function was defined,A test of the adequacy of the function for the obtained regression equation was conducted by performing the F-Snedecor significance test at the assumed confidence level of α = 0.05, verifying the hypotheses on the adequacy of the model by comparing the variance in the approximation errors with the variance in the measurement inaccuracy of the variable determined.

The regression function analysis $S_{e}$ for the machine performance was performed in three stages. The results of the third stage are summarised in Table 2.

As the coefficients $a_{2},a_{4},$ and $a_{5}$ were not significant, the components $a_{2}v_{p}+a_{4}n_{d}+a_{5}h_{d}$ were removed from the general regression equation; the final form of the function describing the dependence of machine efficiency during cutting of onion-type materials on all the independent variables is described by the formula:$$S_{e}=0.7851+0.0475d_{n}-0.051p-1.78\times 10^{-5}v_{p}^{2}-0.0063d_{n}^{2}+0.0038n_{d}^{2}-3.06\times 10^{-5}h_{d}^{2}+0.0067p^{2}$$

The adequacy test yielded the statistic *F* = 587.47, on the basis of which the null hypothesis of inadequacy of the regression function was rejected.

In Table 3, the selected test results obtained during the experiment for $n_{d}$ = 3 pcs. blowing nozzles are summarised.

In Figure 9, Figure 10, Figure 11, Figure 12 and Figure 13, the influence of individual independent variables *v_d_*, *d_n_*, *n_d_*, *h_d_*, and *p* on shaping the effectiveness $S_{e}$ of the onion-peeling machine is presented in a graphic form. The presented relations concern the selected sets of values of the independent variables. In the remaining cases, the trend is the same and the operating efficiency $S_{e}$ assumes higher values, which can be deduced by analysing the course of the multidimensional regression function and the values $S_{e}$ obtained during the experimental studies.

In Figure 9, we present diagrams of the dependence of the machine efficiency $S_{e}$ on the material-feeding speed $v_{p}$. The diagrams unambiguously show that, with an increase in the material feeding speed in the process of cutting the outer layer of the material and cutting off the ends, the effectiveness of the machine’s operation decreases. During the experimental tests, the highest values of the machine’s operating efficiency at a level of $S_{e}$ = 0.78 were obtained for $v_{p}$ = 30 pcs./min and they were on average 7.06% higher than those obtained for *v_p_* = 60 pcs./min. Therefore, the increase in the speed of the onion feeding conveyor above $v_{p}$ = 30 pcs./min results in the decrease in the quality of the realised process.

In Figure 10, there are examples of graphical interpretations of the multidimensional regression function describing the dependence of machine efficiency $S_{e}$ on the cutting depth $d_{n}$ of the external material surface. This dependence is expressed by a polynomial curve of the second degree increasing in the domain of the considered cutting depth values $d_{n}$. The lowest operating efficiency values of $S_{e}$ = 0.645 were obtained for the depth of incision $d_{n}$ = 1.5 mm and they were on average 4.03% lower than the values obtained for $d_{n}$ = 3 mm.

Figure 11 graphically presents the dependence of the machine’s efficiency $S_{e}$ on the number of scale-blowing nozzles $n_{d}$. From the figure, it follows that there is a positive correlation between these two variables. The values of the operating efficiency $S_{e}$ obtained for $n_{d}$ = 4 were on average 6.67% higher than those determined for $n_{d}$ = 2, which indicates that the increase in the effect of the air stream on the onion favours the detachment of cut scales from it.

Figure 12 shows exemplary dependences of the effectiveness of the machine’s operation $S_{e}$ as a function of the distance of nozzles $h_{d}$ blowing off the cut material. In each case of the combination of the remaining variables explaining the function, the dependence is a decreasing function within the range of the values $h_{d}$ of the considered values. The lowest values of operating efficiency $S_{e}$ were obtained for the distance of peeling nozzles from the onion $h_{d}$ of 40 mm and they constituted on average 94.91% of the values obtained for $h_{d}$ = 20 mm.

Figure 13 graphically presents the values of air pressure p supplying the scale-removing nozzles on shaping the efficiency of the machine’s operation $S_{e}$. The curves describing this dependence are the graphs of a second-degree polynomial function increasing in the considered range of variables. The highest values of machine operation efficiency $S_{e}$ corresponded to the pressure p = 7 bar and were, on average, 9.90% higher than those obtained at the pressure p of 4 bar.

## 4. Conclusions

1. The results obtained in the course of the experimental research and the obtained equation of the regression function for the developed machine construction were used to establish systems of independent variables, for which the dependent variable $S_{e}$ reached extreme values.

The values of the independent variables for which the dependent variable $S_{e}$ reached its minimum and maximum values are summarised in Table 4.

2. The effectiveness of the onion-peeling machine’s operation $S_{e}$ increased together with the increase in the depth of the external incisions of the scales $d_{n}$, the number of nozzles blowing the scales $n_{d}$, and the pressure of the air supplying the machine p. Increasing the speed of feeding the material $v_{p}$ and the distance of the air nozzles from the peeled onion $h_{d}$ caused a decrease in the effectiveness of the machine’s operation $S_{e}$.

3. On the basis of the obtained equation of the regression function, it can be concluded that the efficiency of the machine’s operation $S_{e}$ is influenced by all the independent variables considered in the adopted research programme.

4. The application of our globally innovative construction of the machine for peeling onions, based on a modular construction, allowed us to obtain a very high effectiveness of operation at the level of $S_{e}$ = 0.645–0.780, not found in mass-produced machines currently offered on the market

## Figures and Tables

**Figure 1 materials-14-02360-f001:**
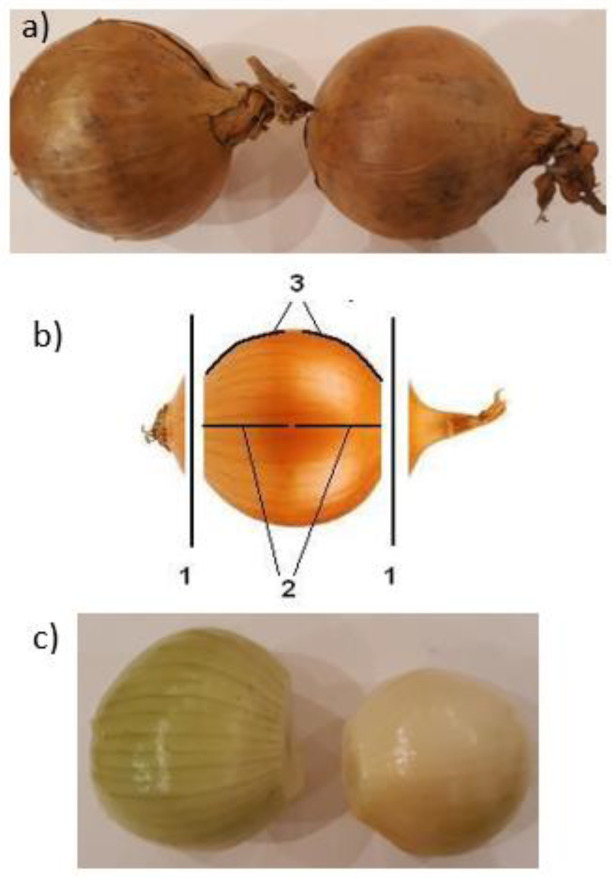
Material subjected to tests: (**a**) onion before the cutting process; (**b**) location of the cutting operations on the onion. 1—places where the ends are cut off by the disc knives, 2—places where the scales are cut by a pair of knives working in a horizontal plane, and 3—places where the scales are cut by a pair of knives working in a vertical plane; (**c**) onions after the cutting process (peeling), and the final product destined for packing.

**Figure 2 materials-14-02360-f002:**
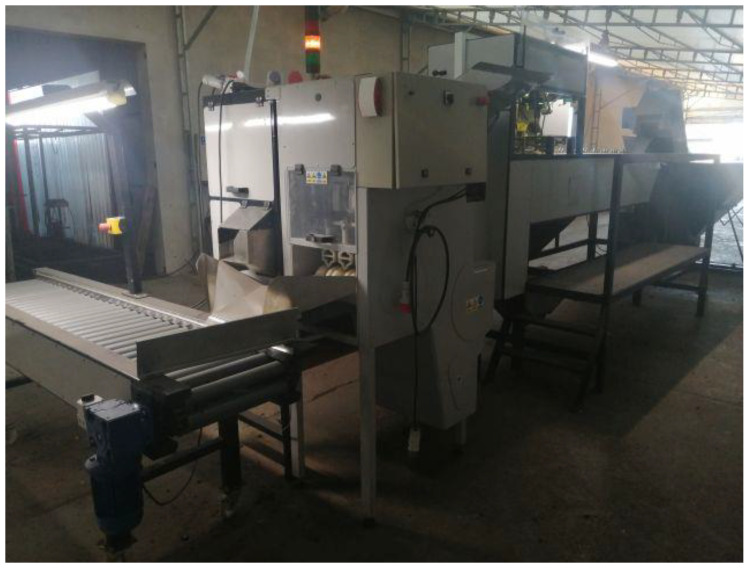
Main test bed: modular machine for peeling onions (left-hand side view).

**Figure 3 materials-14-02360-f003:**
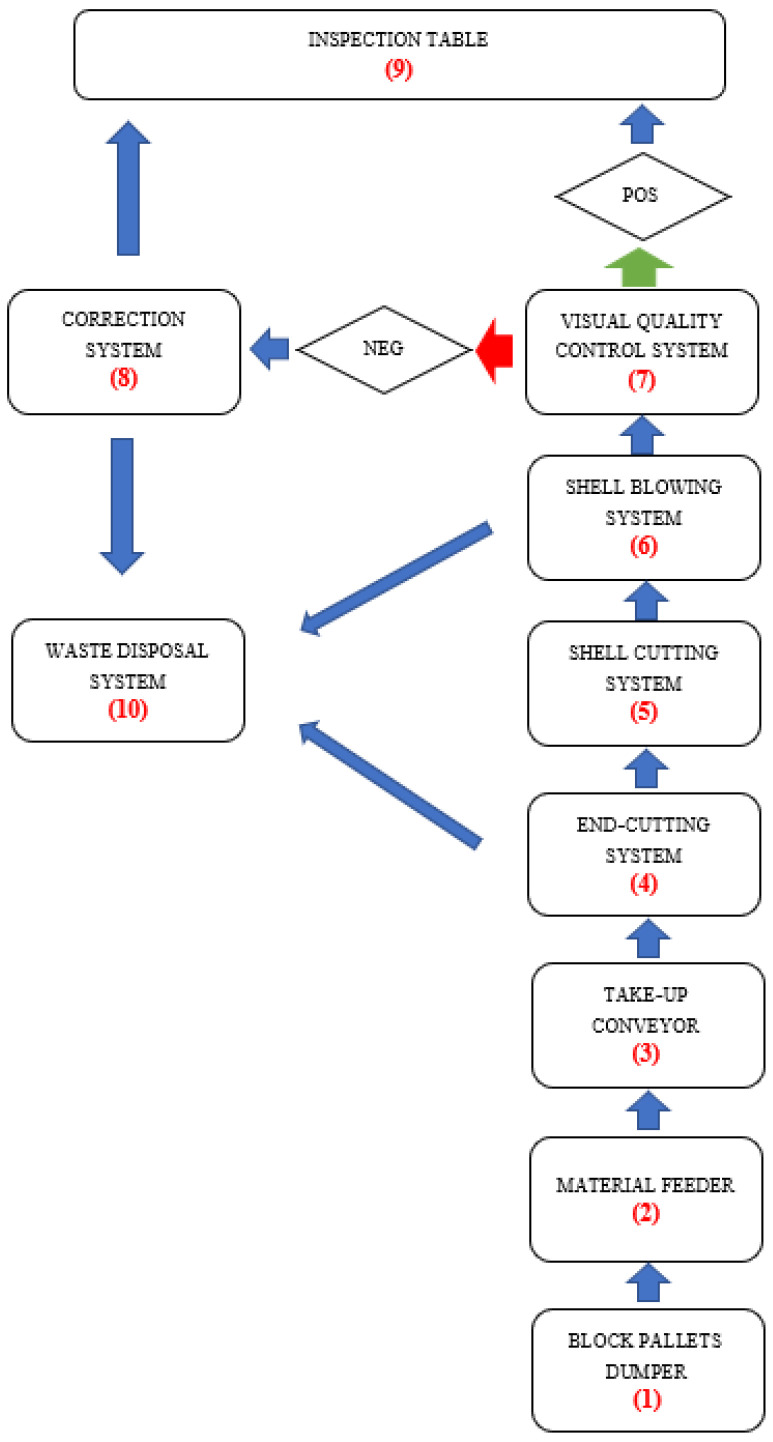
Modular and functional diagram of the test stand construction: POS—positive result, NEG—negative result, (1)—block pallets dumper, (2)—material feeder, (3)—take-up conveyor, (4)—end-cutting system, (5)—shell cutting system, (6)—shell-blowing system, (7)—visual quality control system, (8)—correction system, (9)—inspection table, and (10)—waste disposal system.

**Figure 4 materials-14-02360-f004:**
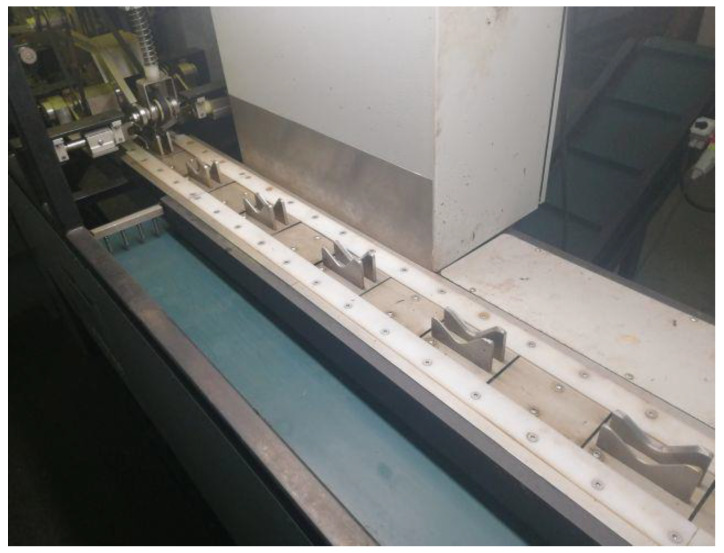
Main take-up conveyor marked with number 3 in Figure 3.

**Figure 5 materials-14-02360-f005:**
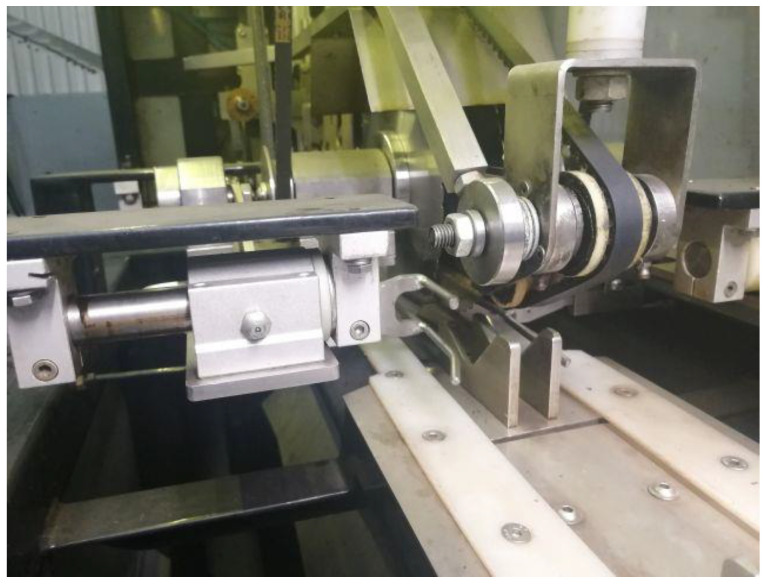
Onion tip cutting system, indicated by the number 4 in Figure 3.

**Figure 6 materials-14-02360-f006:**
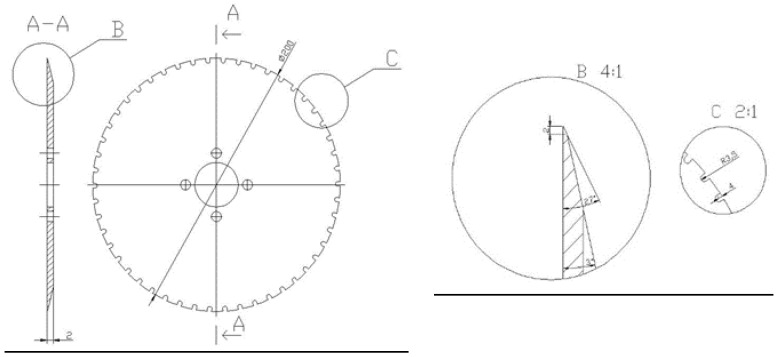
Main geometric dimensions of disc knives for cutting onion ends.

**Figure 7 materials-14-02360-f007:**
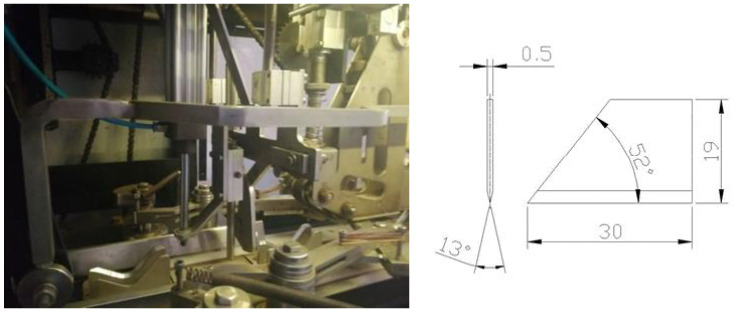
Cutting system of the external surface with the marking of geometrical dimensions of cutting knives (5).

**Figure 8 materials-14-02360-f008:**
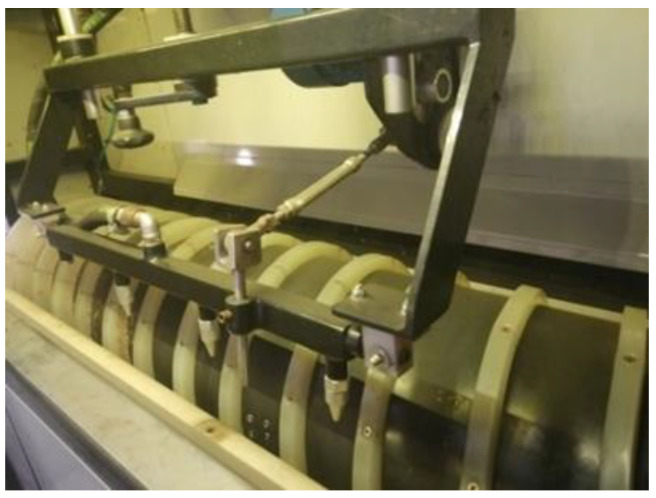
System for blowing off the incised outer layer (scales).

**Figure 9 materials-14-02360-f009:**
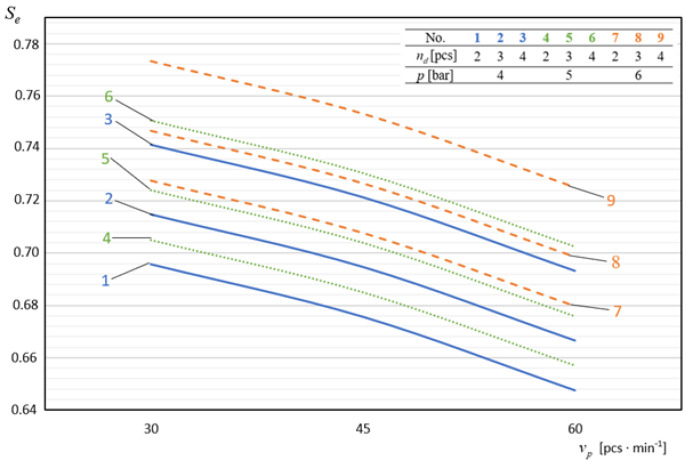
The influence of the material feed rate on shaping the effectiveness of the machine’s operation at constant parameters *d_n_* = 1.5 mm and *h_d_* = 40 mm.

**Figure 10 materials-14-02360-f010:**
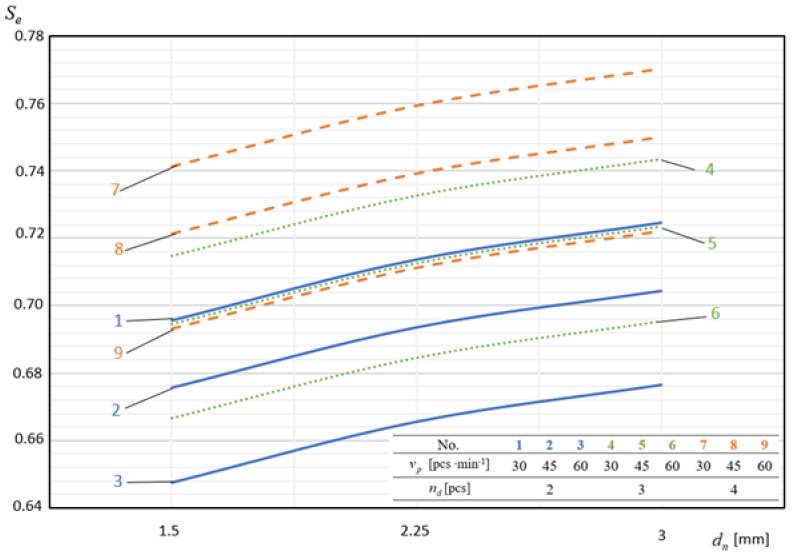
Influence of the depth of incisions of the outer layer of onions on shaping the efficiency of the machine at constant parameters *p* = 4 bar and *h_d_* = 40 mm.

**Figure 11 materials-14-02360-f011:**
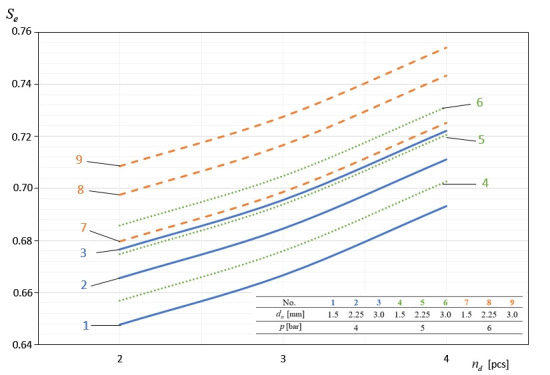
The influence of the number of nozzles removing the incised material (scale) on shaping the efficiency of the machine’s operation at constant parameters *h_d_* = 40 mm and *v_p_* = 60 pcs./min.

**Figure 12 materials-14-02360-f012:**
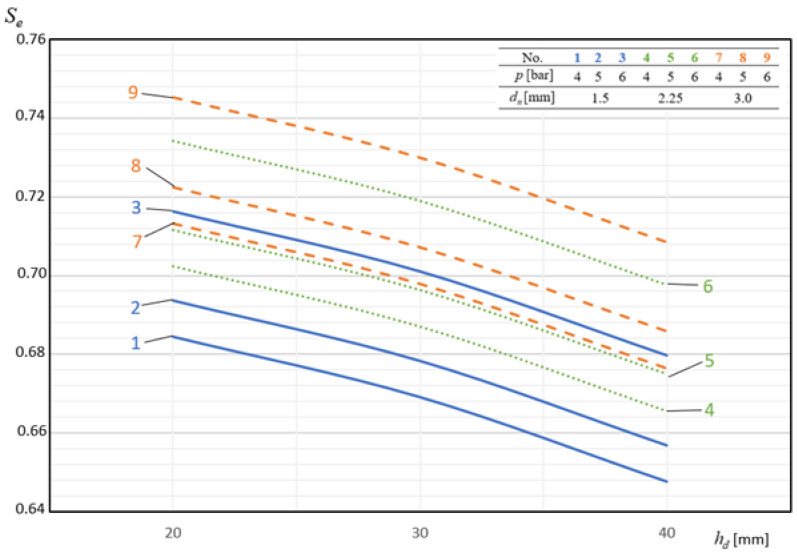
Influence of the distance of the blowing nozzles from the material on the shaping of the machine’s operating efficiency at constant parameters $n_{d}$ = 2 and $v_{p}$ = 60 pcs./min.

**Figure 13 materials-14-02360-f013:**
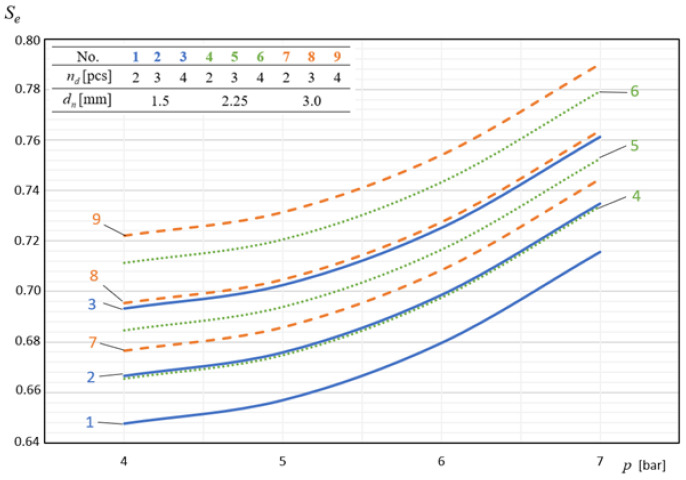
Influence of the air pressure supplying the nozzles removing the scales on the effectiveness of the machine’s operation at constant parameters $v_{p}$ = 60 pcs./min and $d_{n}$ = 1.5 mm. We could not refer to the results of research available in the literature due to the lack of articles on the subjects of onion cutting and their surface layers.

**Table 1 materials-14-02360-t001:** Values of independent variables.

Independent Variables	Values of Independent Variables			
	*x* _1_	*x* _2_	*x* _3_	*x* _4_
Material feeding speed *v_p_* (pcs./min)	30	45	60	
Depth of external shell cuts *d_n_* (mm)	1.5	1.25	3	
Number of air nozzles *n_d_* (pcs.)	2	3	4	
Distance of the air nozzles from the material *h_d_* (mm)	20	30	40	
Air pressure of scale remover *p* (bar)	4	5	6	7

**Table 2 materials-14-02360-t002:** The results of the regression function analysis for the machine performance.

Regression Coefficient	Regression Coefficient Value	Standard Deviation of the Regression Coefficient	Significance Test	
			*t*	*t_kr_*
*a* _1_	0.7851	0.0230	34.13	-
*a* _3_	0.0475	0.0113	4.20	-
*a* _6_	−0.0510	0.0073	6.97	-
*a* _7_	0.0000	0.0000	29.74	1.98
*a* _8_	−0.0063	0.0025	2.51	-
*a* _9_	0.0038	0.0001	28.57	-
*a* _10_	3.06 × 10^−5^	1.35 × 10^−6^	22.73	-
*a* _11_	0.0067	0.0007	10.11	-
Residual standard deviation		0.119	*F*	*F_kr_*
Multidimensional correlation coefficient		0.9636	587.47	1.82

**Table 3 materials-14-02360-t003:** Experimental tests results for *n_d_* = 3 pcs.

$v_{p}\;(\mathrm{pcs}./\min)$	$h_{d}\;(\mathrm{mm})$	p (bar)	$d_{n}\;(\mathrm{mm})$	$S_{e}.\;$	$d_{n}\;(\mathrm{mm})$	$S_{e}$	$d_{n}\;(\mathrm{mm})$	$S_{e}$
30	20	4	1.5	0.7557	2.25	0.7717	3	0.7843
		5		0.7663		0.7825		0.7953
		6		0.7884		0.8051		0.8183
		7		0.8268		0.8443		0.8581
	30	4		0.7462		0.7597		0.7674
		5		0.7566		0.7703		0.7782
		6		0.7785		0.7926		0.8007
		7		0.8164		0.8311		0.8396
	40	4		0.726		0.7345		0.7443
		5		0.7362		0.7448		0.7548
		6		0.7575		0.7663		0.7766
		7		0.7944		0.8036		0.8144
45	20	4	1.5	0.7238	2.25	0.7452	3	0.7575
		5		0.7339		0.7556		0.7681
		6		0.7552		0.7774		0.7903
		7		0.7919		0.8153		0.8287
	30	4		0.7125		0.7335		0.7456
		5		0.7225		0.7438		0.7561
		6		0.7434		0.7653		0.7779
		7		0.7795		0.8026		0.8158
	40	4		0.6911		0.7069		0.7255
		5		0.7008		0.7168		0.7357
		6		0.721		0.7376		0.757
		7		0.7561		0.7734		0.7938
60	20	4	1.5	0.6965	2.25	0.7127	3	0.7252
		5		0.7063		0.7226		0.7353
		6		0.7267		0.7435		0.7566
		7		0.7621		0.7797		0.7934
	30	4		0.6857		0.7015		0.7139
		5		0.6953		0.7114		0.7239
		6		0.7154		0.7319		0.7448
		7		0.7502		0.7675		0.781
	40	4		0.6629		0.6826		0.6924
		5		0.6722		0.6922		0.7021
		6		0.6916		0.7122		0.7224
		7		0.7253		0.7468		0.7575

**Table 4 materials-14-02360-t004:** Layouts of independent variables for the smallest and largest values of the dependent variable $S_{e}$.

Dependent Variable		Independent Variables				
Name	Value	$v_{p}\;(\mathrm{pcs}./\min)$	$d_{n}\;(\mathrm{mm})$	$n_{d}\;(\mathrm{pcs}.)$	$h_{d}\;(\mathrm{mm})$	p (bar)
Effectiveness of machine operation $S_{e}$	min $S_{e}$ = 0.645	60	1.5	2	40	4
	max $S_{e}$ = 0.780	30	3	4	20	7

## Data Availability

All the computational data from mathematical simulations are included in the article.

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
