# Peer review of "Cutting Tests of the Outer Layer of Material Using Onion as an Example"

_materials, 2021, doi:10.3390/ma14092360_

Round 1
Reviewer 1 Report
I am not an expert in the subject treated in this paper, however in my opinion it is not ready to be published in the actual form.
I think that the bibliography review too poor, only 12 papers are cited, and among them at least 5 are self-citations.
At the end of the introduction I would appreciate a small description of the structure of the paper.
In the definition of the effectiveness of the machine I do not understand what does "the mass of all the onions leaving the machine" means.
The bullet list appearing between lines 77 and 84 introduce some parameters that are repeated with more details and the corresponding unities in the next numbered list, the first list must be erased. Moreover, in my opinion, in order to understand correctly the interest of such parameters it would be better to explain before how the machine works, that is, the explanation of the figure 3 should appear before listing the parameters considered.
There is a mistake in the degrees unities in line 125. It is unnecessary to define d in line 129. In line 196 write please Figure instead of drawing.
When considering multivariate regression of second order, equation not numbered, in line 222, why are not considered cross terms like a12 p*v_p and the others?
In line 230 something wrote in Polish. A space missed in line 253 before a_4. In line 268 and in some other lines hereafter S_e is wrote using a different symbol than in line 222, or in table 3.
I think the analysis of polynomial not numbered in line 257 is really poor, and the conclusions are surprising as it is stated that the best effectiveness that could be achieved is 0.780 whereas among the experimental results shown in table 3, there are 26 cases with better results. On the other hand, the worst possible results is stated to be 0.645 while the worst experimental result is 0.663.
Missed the number 1 in the conclusion list.
Author Response
Thank you very much for reading the text of the article carefully and suggesting changes. Undoubtedly, some of the changes will clearly improve the article.
The proposed technical changes are included directly in the text, while other suggestions or doubts are answered here below.
- In response to the comment related to too poor bibliography, we would like to state that the article cites works that in the subjective assessment contribute most to the prepared publication. Nevertheless, the authors are fully aware of the existing publications related to the subject of the article, and the bibliography has been expanded to include the proposed items.
- The presented work is entirely original and in terms of the structure of the article contains the following elements: introduction, which highlights the value of the undertaken work, formulation of the research problem to be solved, description of the research plan and programme, description of the research station equipped with a machine for peeling onions, which was made in the real (production) scale, methodology of the experimental tests, the analysis of the results of the experimental tests and conclusions.
- The effectiveness of the machine is defined as the ratio of the mass of onions properly peeled by the scales blowing unit the mass of all the onions leaving the machine at the research station . The conducted experimental research clearly shows that despite the onions passing through the blower unit, there are some onions which have not been peeled properly and the scales still remain on them.
- The mentioned parameters between lines 77 and 84 are, according to the authors, necessary in the article, because they unambiguously determine: what will be the independent variables in the experimental research and what will be the dependent variables. On the basis of this, the research plan and programme was adopted. The independent variables were clearly defined between lines 92 and 110, while the dependent variable between lines 62 and 66. The onion peeling machine is a rather complex machine in terms of construction and operating principle. Therefore, a detailed description of the design and principle of operation of its particular working units and mechanisms would take several tens of typewritten pages. Therefore, due to the limited volume of the article, the authors decided to include its block diagram, which enables understanding its operation.
- Multiple regression looks for quantitative relationships between the independent variables and the dependent variable. At the stage of conducting it, its output models are adopted, which are further subjected to analysis and verification. In the framework of conducted regression analysis of the research results it turned out that the proposed model (line 222) with rejected component of the form (line 255) describes the relationship between independent variables and the dependent variable at a high value of the multiple correlation coefficient R=0.9636 on the confidence level α=0.05. Therefore it does not seem appropriate to change the form of the output model of the regression function.
- The best machine performance of 0.780 was calculated from a regression equation with a multiple correlation coefficient value of R=0.9636, and is not necessarily the same as the experimental results.
Reviewer 2 Report
The research topic is unusual. At first I thought it was a joke and the work should be rejected. Then I analyzed the problem and found that there is a big demand for peeling onions equipments. There are several methods to achieve this, the method chosen by the authors being one that allows the obtaining of a better quality of the process. The existing equipments on the market and the actual research on this topic should be better analyzed.
I found now some papers which are not in the reference list of the authors:
Development and evaluation of an onion peeling machine El-Ghobashy, H; Adel H. Bahnasawy; Samir A. Ali3; M. T. Afify; Z. Emara
Design And Fabrication Of Automatic Onion Peeling And Cutting Machine P. Ravichandran, C.Anbu, S.Sathish kumar, A.Sakthivel, S.Thenralarasu
https://pdf.directindustry.com/pdf/m-p-engineering/large-onion-peeling-machine/162194-784315.html
So, i ask the authors to improve the state of the art.
The subject is not original. There are many equipments of this kind on the market. The paper does not analyze comparatively the performance of these machines and does not highlight any research need. It's just about optimizing the use of an equipment. This can be a topic for an experimental research.
The paper is easy to read, exception beeing the chapter „ Results”.
How do the authors define the dependent variable „effectiveness of the operation”?
I ask the authors to explain more clearly the experimental method adopted and the analysis of the results.
The bibliography must be reconsidered.
Please specify if the onions are placed in the carriers manually or, the equipment has a poka yoke device that allows the correct positioning of the onions.
At line 230, please translate into English.
Although there is considerable effort behind the paper, I propose a major revision of it.
Author Response
Thank you very much for reading the text of the article carefully and suggesting changes. Undoubtedly some of the changes will clearly improve the article.
The proposed technical changes are included directly in the text, while other suggestions or doubts are answered here below.
- In response to the comment related to too poor bibliography, we would like to state, that the article cites works, which in the subjective assessment contribute most to the prepared publication. Nevertheless, the authors are fully aware of the existing publications related to the subject of the article and the bibliography has been extended by the suggested positions.
- The article does not analyse the performance of existing onion peeling machines, because their forms of construction and operating parameters are significantly different, so it would not bring much new to science. It would only be possible to make a popular-scientific study (article), which was not the intention of the authors.
- The effectiveness of the machine is defined as the ratio of the mass of onions properly peeled by the scales blower to the mass of all onions leaving the machine at the research station . he conducted experimental research clearly shows that in spite of passing the onions through the blower unit, there are some onions which have not been peeled properly and the scales still remain on them.
- Onions from the in-feed hopper are picked up directly by the take-up conveyor and transported to the end-cutting system. The design of the take-up conveyor forces the onion to be positioned correctly
Round 2
Reviewer 2 Report
Dear authors,
When presenting the current state of a field, it is necessary to detail the most important aspects. In your case, listing two analogous studies is not enough.
You should specify what pelling onion systems exist. What manufacturers of such equipment exist on the market. What are the advantages and disadvantages of each method.
For each previous study performed, the author must be specified, what type of procedure he used, what is the efficiency of the procedure, etc. These things are not detailed in your work. Then you should specify the benefits of your equipment. Only then can you begin to describe the onion peeling method you use.
You also did not specify whether loading the onion in the correct position is done manually or automatically.
Please do these improvements.
Author Response
Thank you very much for the quick processing of the article.
The changes suggested by the reviewer were applied directly to the text of the article. The most important items of changes are marked in red.